# Serine Rejuvenated Degenerated *Volvariella volvacea* by Enhancing ROS Scavenging Ability and Mitochondrial Function

**DOI:** 10.3390/jof10080540

**Published:** 2024-08-01

**Authors:** Qiaoli Wang, Jianing Zhu, Yonghui Wang, Jianmin Yun, Yubin Zhang, Fengyun Zhao

**Affiliations:** 1College of Food Science and Engineering, Gansu Agricultural University, Lanzhou 730070, China; 18409483023@163.com (Q.W.); 18152275316@163.com (J.Z.); 18298762316@163.com (Y.W.); yunjianmin@gsau.edu.cn (J.Y.); zhangyb@gsau.edu.cn (Y.Z.); 2Kangle County Special Agricultural Development Center, Linxia 731599, China

**Keywords:** *Volvariella volvacea*, serine, strain rejuvenation, antioxidase, lignocellulase, mitochondria

## Abstract

Serine is a functional amino acid that effectively regulates the physiological functions of an organism. This study investigates the effects of adding exogenous serine to a culture medium to explore a feasible method for the rejuvenation of *V. volvacea* degenerated strains. The tissue isolation subcultured strains T6, T12, and T19 of *V. volvacea* were used as test strains, and the commercially cultivated strain V844 (T0) was used as a control. The results revealed that the addition of serine had no significant effect on non-degenerated strains T0 and T6, but could effectively restore the production characteristics of degenerated strains T12 and T19. Serine increased the biological efficiency of T12 and even helped the severely degenerated T19 to regrow its fruiting body. Moreover, exogenous serine up-regulated the expression of some antioxidant enzyme genes, improved antioxidase activity, reduced the accumulation of reactive oxygen species (ROS), lowered malondialdehyde (MDA) content, and restored mitochondrial membrane potential (MMP) and mitochondrial morphology. Meanwhile, serine treatment increased lignocellulase and mycelial energy levels. These findings form a theoretical basis and technical support for the rejuvenation of *V. volvacea* degenerated strains and other edible fungi.

## 1. Introduction

Edible mushrooms are a vital source of natural food and medicinal products [1]. The quality and yield of edible mushrooms are significantly impacted by the strains, and strain degeneration is a common issue for numerous edible mushrooms. *Volvariella volvacea* (*V. volvacea*) is an important cultivated edible mushroom in China, accounting for more than 80% of the world’s total production [2]. However, according to data from the China Edible Mushroom Association (CEMA), the annual production of *V. volvacea* has shown a consistent decrease over the past decade, with strain degeneration being a significant contributing factor.

The low-temperature preservation method has proven to be an effective strategy to decelerate mycelial growth and reduce the occurrence of genetic mutations [3]. Nevertheless, *V. volvacea*, characterized by its preference for high temperatures, does not tolerate low-temperature storage conditions. In fact, the mycelial of *V. volvacea* autolyzes and perishes within 48 h at 4 °C [4]. As a result, compared to other edible mushrooms, *V. volvacea* experiences more frequent strain degeneration. This highlights the pressing need for an effective rejuvenation method for this mushroom.

Amino acids play an indispensable role in the sustenance of living organisms and are tightly connected to life processes. They serve as an organic nitrogen source, promoting the growth of edible mushrooms, and function as essential regulators of organismal activities [5]. During the initial stages, our research group rejuvenated the degenerated strains of *V. volvacea* by introducing 20 distinct amino acids, and the results revealed that serine had a remarkable capacity to enhance the mycelial characteristics of degenerated *V. volvacea* strains [6]. Serine, as the fundamental building block of proteins, not only participates in various metabolic processes but also exerts its influence through metabolites [7]. Moreover, serine plays a pivotal role in the construction and maintenance of cell membranes, the synthesis of muscle tissue, and the sheaths surrounding nerve cells [8]. Garcia et al. discovered that the addition of serine to a nutrient solution significantly increased the mineral content of plant leaves [9]. Likewise, Zhang et al. observed a growth-promoting effect of serine on tomato seedlings when serine was supplemented with a hydroponic solution [10].

In this study, the *V. volvacea* subcultured strains T6, T12, and T19 obtained by tissue isolation were used as test strains, and the commercially cultivated strain V844 (T0) was used as the control strain. Serine was added to the culture medium (potato dextrose agar, PDA) and cultivation substrate to investigate its effects on production characteristics and ROS scavenging ability of *V. volvacea* subcultured strains. This study provides a theoretical basis and technical support for the rejuvenation of *V. volvacea* and other edible fungi degenerated strains.

## 2. Materials and Methods

### 2.1. Strains and Media

The original strain of *V. volvacea* (T0), a commercialized cultivated strain V844, was conserved in the College of Food Science and Engineering, Gansu Agricultural University, China.

*V. volvacea* subcultured strains (T6, T12, and T19) were obtained through the successive tissue isolation subculturing method. Briefly, T0 was cultivated, and an egg-shaped stage fruiting body was obtained. The fruiting body was cut, and a small piece from the junction of the stipe and the cap was used to generate the 1st generation strain, labeled as T1. T1 was cultivated, and the obtained fruiting body was used to obtain the 2nd generation strain, labeled as T2. Likewise, T1-T19 strains were obtained using 19 consecutive cultivations (Appendix A). T6, T12, and T19 were selected as experimental strains.

The media for strain cultivation were as follows: potato dextrose agar (PDA) medium (*w*/*w*) contained potato 20.0%, glucose 2.0%, KH_2_PO_4_ 0.1%, MgSO_4_ 0.1%, and agar powder 2.0%. The seed culture medium (*w*/*w*) contained cottonseed hull 88%, bran 10%, gypsum 1%, lime 1%, and water content 65%. The cultivation medium (*w*/*w*) included waste cotton 97%, lime 3%, and water content 70%. For serine treatments, the test water in PDA medium, seed medium, and cultivation medium were all replaced with 0.2% aqueous serine solution. Serine (Shanghai Yuanye Biotechnology Co., Ltd., Shanghai, China) was L-serine, and the water used in the PDA medium, seed medium, and cultivation medium was distilled water.

### 2.2. Observation of Colonies Morphology

All strains were inoculated onto PDA medium and incubated at 30 °C for 72 h. Colony morphology was photographed and observed using the method of Ko et al. [11].

### 2.3. Determination of Mycelial Biomass

Mycelial biomass was determined by referring to the method of Scheid et al. [12]. The cellophane was cut into a size slightly smaller than the petri dish, and after sterilization, the sterile cellophane was covered on the surface of the solidified PDA plate. Strains were inoculated with 1 piece of mycelial tips-containing agar block (6 mm diameter punch) on a PDA plate at 30 °C for 72 h. The mycelial was then gently scraped off and weighed, and the data were recorded.

### 2.4. Determination of Mycelial Branching

Mycelial branching was determined by referring to the method of Guo et al. [13]. A piece of mycelial block was put into the center of a PDA plate, and clean and sterilized coverslips were inserted into the plate at a 45° angle. The plate was labeled, sealed, and incubated at 30 °C for 72 h. Afterward, the branching of the mycelial was observed under a light microscope (Carl Zeiss, Oberkochen, Germany), and the number and branching of mycelial in the same field of view were counted and photographed.

### 2.5. Cultivation Test

Referring to the method of Li et al. [14], the cultivation test of *V. volvacea* was carried out. Due to the complex composition of the cultivation substrate, it was thoroughly mixed and soaked overnight with sufficient distilled water. On the following day, excess water was removed from the cultivation medium, which was subsequently weighed, distributed into 750 mg packets, and left overnight. Finally, on the third day, sterilization was carried out at 121 °C for 210 min. Plastic frames measuring 40 cm × 20 cm × 10 cm were each loaded with 1.2 kg of the sterilized matrix substrate. The cultivation baskets were covered with plastic bags and placed at a constant temperature of 30 °C (Shanghai-Heng Scientific Instrument Co., Ltd., Shanghai, China). Once the mycelial grew all over the frame, enough water was sprayed, and the lighting and humidifier were turned on to raise the temperature to 33 °C at a relative humidity of 85–90%. Three replicates were set for each strain to calculate the cultivation indices.

The production cycle (days) represents the time needed for the fruiting body to grow into the harvesting stage (egg-shaped stage) after cultivation. Biological efficiency (%) was calculated as follows:(1)$$Biological\;efficiency\left(\%\right)=\frac{\left(fresh\;fruiting\;body\;yield\right)}{\left(quantity\;of\;dry\;substrate\;used\right)}\times 100\%$$

### 2.6. Estimation of Antioxidant Enzyme Gene Expressions by Real-Time Quantitative PCR (RT-qPCR)

Referring to the genome sequence of *V. volvacea* (http://jgi.doe.gov, accessed on 5 November 2022), primers were designed using Primer Premier 5.0 software. The endosomal SPRY domain protein (SPRYp) gene was used as the internal reference gene, and the primer sequences are listed in Appendix A. Referring to the method of Lei et al. [15], RT-qPCR amplification was performed, and the relative gene expression was calculated using the 2^−ΔΔCt^ method.

### 2.7. Determination of Antioxidant Enzyme Activity and ROS Content

Based on the method of Guo et al. [16], the strains were inoculated in PDA media for 3 days. The obtained mycelial was used to determine intracellular levels of superoxide anion (O_2_^−^), hydrogen peroxide (H_2_O_2_), and activity of peroxidase (POD), superoxide dismutase (SOD), catalase (CAT), ascorbate peroxidase (APX), glutathione peroxidase (GPX) glutathione reductase (GR) using respective commercial kits (Beijing Solepol Science and Technology Co., Ltd., Beijing, China) following manufacturers’ instructions.

### 2.8. Nitrouble Tetrazolium Chloride (NBT) Staining

NBT (Shanghai Blue Season Technology Development Co., Ltd., Shanghai, China) staining was performed following the method of Parkhey et al. [17]. The method of slide preparation for mycelial was similar to Section 2.4. When the mycelial grew to about half of the slide, the slide was removed and added with 20 μL of 0.3 mmol/L NBT solution onto the mycelial densely populated area. The staining was performed at room temperature for 15 min, and then the slide was observed under a light microscope.

### 2.9. Mitochondrial Staining

Mitochondrial staining was performed according to Xu et al. [18]. The slide preparation method for mycelial was similar to Section 2.4. When the mycelial grew to about 1/3 of the slide, 3.7% paraformaldehyde (Shanghai Yuanye Biotechnology Co., Ltd., Shanghai, China) was added dropwise to the mycelial-grown area. After 20 min, the slide was rinsed with phosphate buffer saline (Shanghai Yuanye Biotechnology Co., Ltd., Shanghai, China) (PBS) 2–3 times and observed under an inverted fluorescence microscope.

### 2.10. Measurement of Mitochondrial Membrane Potential (MMP)

MMP was determined as described by Moghadasi et al. [19]. The method of mycelial slide preparation was similar to Section 2.4. When the confluency reached about 1/3 of the slide, it was added with 3.7% paraformaldehyde. After 20 min, slides were washed with PBS 3 times, followed by addition with the Mito-Tracker Red CMXRos (Shanghai Biyuntian Biotechnology Co., Ltd., Shanghai, China). The incubation was performed for 20 min, followed by washing 3 times with PBS. Finally, the slides were observed under an inverted fluorescence microscope (Carl Zeiss, Oberkochen, Germany).

### 2.11. Determination of Strain Energy

Strain energy was determined by referring to the method of Peng et al. [20]. First, 0.2 g of mycelial was weighed, ground in an ice bath, and then added with 2 mL of 3% perchloric acid solution. The mixtures were centrifuged at 10,000 rpm for 10 min at 4 °C. From the obtained supernatants, 1 mL was added with 60 μL of 20% KOH solution, and the mixture was allowed to stand for 20 min at 4 °C. Afterward, the mixture was again centrifuged at 10,000 rpm for 10 min at 4 °C The obtained supernatants were stored at −30 °C.

Accurately weighed 5′-adenosine triphosphate (ATP), 5′-adenosine diphosphate (ADP), and 5′-adenosine monophosphate (Shanghai Yuanye Biotechnology Co., Ltd., Shanghai, China) (AMP) 20 mg standards were dissolved in ultrapure water and then diluted into 7 standard solutions of 5, 10, 50, 100, 200, 300, and 400 μg/mL in 20 mL volumetric flasks, respectively.

The standards and sample solutions were filtered through 0.45 μm polytetrafluoroethylene membranes and analyzed on an Agilent 1100 high-performance liquid chromatography (HPLC) (Agilent Technology Co., Ltd., Guangdong, China) system equipped with an Eclipse Plus C18 column (5 μm, 150 × 4.6 mm). The chromatographic conditions were as follows: column temperature, 30 °C; mobile phase, a mixture of phosphate buffer (0.1 mol/L) and methanol (99.9:0.1) at pH 4.1; flow rate, 1 mL/min; flow time, 10 min; detection wavelength, 282 nm; injection volume, 20 μL. All tests had three replicates, and the calculation was performed based on peak area. The energy charge (EC) was calculated as follows:(2)$$EC=\frac{ATP+1/2ADP}{ATP+ADP+AMP}$$

### 2.12. Determination of Malondialdehyde Content (MDA)

MDA content was determined by referring to the method of Pan et al. [21]. 0.5 g mycelial was added with 5 mL of 5% trichloroacetic acid (TCA) and grounded. The resulting homogenate was centrifuged at 3000 r/min for 10 min. Subsequently, 2 mL of the obtained supernatant was added to 0.67% thiobarbituric acid (Shanghai Yuanye Biotechnology Co., Ltd., Shanghai, China) (TBA) 2 mL, and the mixture was boiled for 30 min in a water bath at 100 °C. Then, the mixture was centrifuged again after cooling. The absorbance values of the supernatant at 450, 532, and 600 nm were determined, respectively, and the MDA concentration was calculated according to the formula. The MDA content (μmol/g) in the fresh-weight tissue was also calculated.
(3)$$MDA\;content=\frac{[6.45({OD}_{532}-{OD}_{600})-0.56\times {OD}_{450}]\times total\;sample\;volume}{sample\;mass}$$

### 2.13. Determination of Matrix-Degrading Enzyme Activity

According to the method of Qin et al. [22], *V. volvacea* mycelial was incubated at 33 °C with 180 rpm shaking for 8 days. An appropriate amount of fermentation broth was taken out and centrifuged at 10,000 rpm at 4 °C for 10 min. The obtained supernatant was considered the crude enzyme solution. Then filter paper enzyme (FPA), endoglucanase (EG), exoglucanase (CBH), β-glucosidase (BGL), laccase (Lac), manganese peroxidase (MnP), hemicellulase (HMC) and xylanase (Xyl) activities were determined using respective commercial kits (Beijing Solepol Science and Technology Co., Ltd., Beijing, China) according to the manufacturer’s instructions.

### 2.14. Data Processing

The determination of all indicators was repeated three times for each strain to obtain average values. The data were statistically processed and plotted using Microsoft Excel 2010 software (Microsoft, Redmond, WA, USA), analyzed by one-way ANOVA using SPSS 19.0 (SPSS, Chicago, IL, USA), and the correlation was illustrated using Origin 2021 software (Electronic Arts Inc., Chicago, IL, USA).

## 3. Results

### 3.1. Effect of Serine on Mycelial Characteristics of V. volvacea

Strains T0, T6, T12, and T19 were uniformly cultured for 3 days to determine their colony morphology and mycelial biomass. The results revealed that with increasing subculture, both the colony diameter and mycelial biomass first increased and then decreased. After the exogenous addition of serine in the PDA medium, the colony diameters of T0, T6, T12, and T19 strains increased to different degrees, and the mycelial biomass increased by 1.90%, 2.21%, 8.31%, and 11.19%, respectively, compared to the control group (Figure 1A,B). Importantly, the mycelial biomass of serine-treated T12 and T19 strains was significantly higher than that of the respective control groups (*p* < 0.05).

Microscopic observation of *V. volvacea* mycelia under the same magnification showed that with the increasing subculture, mycelial branching reduced and the branching rates decreased. After the exogenous addition of serine, the mycelial branching of T6, T12, and T19 strains significantly improved compared with control groups, while there was no effect on T0 (Figure 2A,B).

### 3.2. Effect of Serine on Fruiting Body Characteristics of V. volvacea

T0, T6, T12, and T19 strains were cultivated, and the results are shown in Figure 3. T0, T6, and T12 produced fruiting bodies, while T19 could only form primordia, which did not grow into fruiting bodies. The number of fruiting bodies of T0, T6, and T12 strains showed different degrees of increase after exogenous addition of serine treatment. Intriguingly, serine restored the fruiting ability of T19.

The production cycle and biological efficiency were further determined and the results are shown in Figure 4. There was no significant difference between the production traits of T6 and T0. However, compared with T0, the biological efficiency of T12 decreased by 49.02%, and the production cycle increased by 5.88%. The exogenous addition of serine had no significant effect on the production traits of T0 and T6. However, compared with the control group, serine addition increased the biological efficiency of T12 by 27.47% and decreased the production cycle by 5.88%.

### 3.3. Effect of Serine on the Relative Expression of Antioxidant Enzyme Genes in V. volvacea

The genome database of *V. volvacea* was found to have three sod genes (*Cu/Zn-sod*, *Mn-sod1*, *Mn-sod2*), two cat genes (*cat1*, *cat2*), one gr gene, and one gpx gene. The results of the RT-qPCR analysis of seven antioxidant enzyme-related genes are shown in Figure 5. With the increase in the number of successions, the relative expression of the *Mn-sod1* gene did not change significantly, the relative expression of the *cat2* gene decreased gradually, and the relative expression of *Cu/Zn-sod*, *Mn-sod2*, *cat1*, *gpx*, and *gr* genes first showed an increasing trend followed by a decline. After serine treatment, the relative expression of *Cu/Zn-sod*, *Mn-sod2*, and *gpx* genes did not change significantly in T0 and T6 but increased significantly in T12 and T19 (*p* < 0.05). In T12 and T19, the expression of the *Cu/Zn-sod* gene was increased by 20.51% and 54.91%, the *Mn-sod2* gene was increased by 28.88% and 19.04%, and the gpx gene was increased by 19.85% and 35.57%, respectively, compared with the control group. The relative expression of *Mn-sod1*, *cat1*, *cat2*, and *gr* genes showed insignificant changes in relative expression compared with the control group (*p* > 0.05).

### 3.4. Effect of Serine on the Antioxidase Activity of V. volvacea

The changes in antioxidant enzyme activities of T0, T6, T12, and T19 are shown in Figure 6. POD, GPX, SOD, CAT, APX, and GR activities decreased with the increasing subculture. Exogenous addition of serine significantly increased the activities of POD, GPX, and SOD in T0, T6, T12, and T19 strains (*p* < 0.05); they increased in T0 by 6.98%, 10.14%, and 4.70%, in T6 increased by 13.51%, 10.58% and 8.15%, in T12 increased 22.22%, 38.79% and 26.20%, and in T19 increased by 46.00%, 92.77% and 60.05%, respectively. Exogenous serine significantly increased the CAT activity of T6, T12, and T19 by 50.03%, 76.05%, and 166.47%, and significantly increased the APX activity of T12 and T19 by 10.52% and 31.14%, respectively.

### 3.5. Effect of Serine on ROS Content in V. volvacea

T0, T6, T12, and T19 mycelia were stained with NBT and observed microscopically (Figure 7A). In control groups, the color of the mycelial gradually darkened with the increasing subculture; the color of the T19 mycelial was the darkest, followed by T12. After the serine treatment, T0 mycelia showed no effect on color change, while the color of T6 became significantly lighter. Meanwhile, the colors of T12 and T19 also became lighter but did not revert to the color of T0. With increasing subculture, the contents of both H_2_O_2_ and O_2_^−^ showed an increasing trend in the mycelia. Exogenous addition of serine significantly decreased O_2_^−^ content by 8.14%, 15.22%, and 22.19% (Figure 7B), and H_2_O_2_ content by 24.44%, 14.98%, and 36.99%, respectively (Figure 7C).

### 3.6. Effect of Serine on Mitochondria of V. volvacea

T0, T6, T12, and T19 were stained with the mitochondria-specific fluorescent dye Mito-Tracker Green and observed under an inverted fluorescence microscope (Figure 8A). There were more bright spots in the T0 and T6 strains, while such spots were significantly reduced in T12 and T19. After exogenous serine treatment, no significant change occurred in the bright spots of T0 and T6, while the bright spots significantly increased in T12 and T19.

Furthermore, the strains were stained with MMP-dependent fluorescent dye and observed under an inverted fluorescence microscope (Figure 8B). In the control group, the fluorescence intensity of the mycelial gradually weakened with the increasing subculture. The fluorescence intensity of T0 and T6 did not change significantly, while it significantly weakened in T12 and T19. The exogenous addition of serine did not significantly change the fluorescence intensity in T0 and T6, while it significantly enhanced the fluorescence intensity in T12 and T19. These results indicated that exogenous serine treatment recovered the mitochondrial morphology of degenerated strains (T12 and T19).

### 3.7. Effect of Serine on Mycelial Energy of V. volvacea

The energy changes in the mycelial of *V. volvacea* somewhat reflect the vigor of mitochondria. With the increasing successions, the ATP and ADP contents showed a decreasing trend, while the AMP and EC values did not change significantly. Exogenous serine addition increased ATP and ADP contents in T12 and T19, in which ATP contents in T12 and T19 increased by 18.50% and 14.75% (Figure 9A), and ADP contents increased by 45.20% and 43.86% (Figure 9B), respectively. ATP and ADP contents of T0 and T6 did not change significantly. Exogenous serine significantly increased AMP content in T0 and T6 by 11.74% and 4.99%, respectively (Figure 9C). The EC value did not change significantly (Figure 9D).

### 3.8. Effect of Serine on MDA Content of V. volvacea

The MDA content of *V. volvacea* mycelial was determined, and the results are shown in Figure 10. MDA content increased gradually with the increase of subgenerations. After serine treatment, the MDA content of T0, T6, T12, and T19 decreased. The MDA content of T0 and T6 was not significantly different from that of the control group, but it significantly decreased in T12 and T19 by 50.17% and 37.90%, respectively (*p* < 0.05).

### 3.9. Effect of Serine on the Matrix-Degrading Enzymes of V. volvacea

The cellulose degradation-related enzyme activities of T0, T6, T12, and T19 were determined (Figure 11). The FPA, BGL, and EG enzyme activities significantly decreased with the increasing successions, while the CBH enzyme activity did not show a clear change. After the exogenous addition of serine, the FPA and EG activities of T6, T12, and T19 were significantly increased (*p* < 0.05). FPA and EG activities of T6 were increased by 13.17% and 9.52%, T12 by 20.75% and 10.7%, and T19 by 30.01% and 18.78%, respectively. The BGL activities of T12 and T19 increased significantly by 12.12% and 20.18%, respectively. The CBH activity of T6 was significantly increased by 41.73% (*p* < 0.05).

The results of lignin degradation-related enzymes showed that with the increasing subculture, Lac activity first increased and then decreased, while the activity of MnP decreased. Exogenous addition of serine significantly increased the Lac activities of T6, T12, and T19 by 41.82%, 33.33%, and 47.14%, respectively (*p* < 0.05). MnP activities of T12 and T19 were significantly elevated by 72.34% and 106.67%, respectively (Figure 11).

The results of hemicellulose enzymes showed that HMC activity first increased and then decreased with the increasing subculture, while the Xyl activity did not change significantly (Figure 11). Compared with the control group, the exogenous addition of serine significantly increased the HMC and Xyl activities in T6 by 7.80% and 4.10%, respectively (*p* < 0.05).

### 3.10. Correlation between Main Characteristics of V. volvacea

The results of the correlational analysis are summarized in Figure 12. ROS (O_2_^−^, H_2_O_2_) were negatively correlated with antioxidant enzymes (SOD, CAT, GPX). Mycelial biomass and biological efficiency were positively correlated with antioxidant enzymes (SOD, CAT, GPX), matrix-degrading enzymes (CBH, Lac, Xyl), mycelial energy (ATP, EC value), and negatively correlated with ROS (O_2_^−^, H_2_O_2_) and MDA content. Production cycles were positively correlated with ROS (O_2_^−^, H_2_O_2_) and MDA content, while they negatively correlated with antioxidant enzymes (SOD, CAT, GPX), matrix-degrading enzymes (CBH, Lac, Xyl), mycelial energy (ATP, EC value).

## 4. Discussion

The breeding of edible fungi requires significant time and effort. Breeders invest substantial time and resources in breeding excellent varieties. However, strain degradation may occur after 1–2 years or even after several subgenerations. These degenerated strains lose their original good traits and their production capacity decreases, which causes huge economic losses to the producers [23]. Adding exogenous nutrients can stimulate the growth of mycelia and increase the yield of fruiting bodies, which is an important means for the rejuvenation of degraded strains [24]. Hou et al. enhanced the fruiting body number and biological efficiency of *V. volvacea* by adding sodium acetate to the cultivation substrate [25]. Sardar et al. improved the yield and nutrition of king oyster mushrooms by adding moringa leaf powder to cotton waste [26]. Ren et al. observed that the addition of Na_2_SeO_3_ to the medium led to thicker mycelial and increased biomass of *Cordyceps militaris* [27]. Cao et al. found that under salt, dehydration, or cold stress, treatment with 5 mM GABA significantly enhanced the mycelial growth rate and density of both *H.marmoreus* strains by promoting front hyphae branching [28]. Our study found that the addition of serine to the culture medium had no significant effect on the mycelial and fruiting body indicators of both T0 and T6 strains. Exogenous serine significantly increased the colony diameter, mycelial biomass, and mycelial branching of T12 and T19 (Figure 1 and Figure 2), shortened the production cycle, and significantly improved the biological efficiency of T12, restoring it to the level of T0; even the T19 strain, which had lost fruiting ability, was capable of regrowing fruiting bodies (Figure 3). Serine made the T19 strain regrow, which had otherwise lost the ability to produce fruiting bodies. However, the production cannot be restored to the level of the original species. Since this was a laboratory study, whether serine can improve the production traits of degraded *V. volvacea* in actual production needs to be further investigated.

The NBT chemical staining method allows qualitative analysis of ROS content. The ROS production sites appear bluish-purple and colorless in case of no production [29]. Shinogi et al. stained Japanese pear shoots by the NBT method and revealed that ROS-producing shoots showed a blue color [30]. Likewise, Javvaji et al. found that NBT reacted with intracellular ROS to produce a blue-violet substance [31]. Ma et al. carried out NBT staining experiments on *Metarhizium anisopliae* mycelial and showed that the normal mycelial showed a light purple color after NBT staining, while the degraded mycelial were of dark blue-purple color [32]. These results showed that the color depth of NBT staining positively correlates with the amount of ROS content, and the strain degradation is closely related to the accumulation of ROS. In this study, we showed that after NBT staining, as the degradation of *V. volvacea* increased, the color of the mycelial became darker with the increase in ROS content. However, after the exogenous addition of serine, the mycelial color of degenerated strains became lighter, and the ROS content reduced (Figure 7). This result indicated that the exogenous addition of serine decreased the ROS content in the mycelial.

Although fungal degeneration has long been recognized, the underlying causes remain poorly understood. Wang et al. [33] were the first to propose that the degeneration of filamentous fungi may be a sign of aging, linked to oxidative damage caused by the accumulation of excessive ROS in degenerated strains [34]. This finding is consistent with our finding that the ROS content in *V. volvacea* gradually increases with increasing generations (Figure 7). Serine, a crucial amino acid in organisms, impacts various signaling and biosynthetic pathways, including the synthesis of phospholipids, purines, pyrimidines, and more [35]. Moreover, serine plays a pivotal role as a one-carbon donor in the folate cycle, contributing to nucleotide synthesis, methylation reactions, and NADPH production for antioxidant defense [36]. Furthermore, serine serves as a precursor for the synthesis of glycine and cysteine, both of which ultimately contribute to the production of glutathione, a vital antioxidant. Gao et al. demonstrated that the proliferation and differentiation of HCT116 cells were inhibited in a serine-free medium compared to a serine-containing medium [37]. Xiong et al. successfully restored the fruiting ability of degenerated strains of *Cordyceps militaris* by enhancing their antioxidant capacity [38]. Zhang et al. found that exogenous melatonin inhibited darkness-induced senescence of perennial ryegrass leaves by scavenging ROS through the activation of the superoxide dismutase-peroxidase antioxidant pathway [39]. Our results showed that serine increased the expression of *Cu/Zn-sod*, *Mn-sod2*, and *gpx* genes in degenerated strains T12 and T19 (Figure 5), increased SOD, GPX viability (Figure 6), decreased ROS accumulation (Figure 7), and improved the production traits of degraded strains (Figure 3 and Figure 4). Correlation analysis also showed that SOD, CAT, and GPX activities were positively correlated with biological efficiency, while ROS (O_2_^−^, H_2_O_2_) content was negatively correlated with biological efficiency. In this paper, the relative expression of only eight antioxidant enzyme-related genes and the changes in the activities of four antioxidant enzymes were determined after the exogenous addition of serine. The specific regulatory mechanisms of antioxidant enzyme-related genes must be further investigated.

Mitochondria serve as the primary site of ROS production in the cell, but excessive ROS-induced oxidative stress can damage mitochondria, resulting in even more ROS production and further mitochondrial damage. This forms a vicious circle, which is the theoretical basis for the “mitochondria cause aging” theory [40]. The mitochondrial membrane potential reflects the structural integrity of mitochondria, which subsequently impacts the mitochondrial respiratory chain electron transfer, inner and outer membrane integrity, substance transport, and mitochondrial morphology [41]. Berry et al. showed that an optogenetic approach that uses light-activated proton pumps to increase MMP in adulthood can improve age-related phenotypes and extend the lifespan of elegant mice [42]. Li et al.observed a significant decrease in MMP in mycelial cells of degenerated strains of *Streptomyces griseus* [34]. In this study, it was observed that the mitochondrial membrane potential of degenerated strains of *V. volvacea* decreased, and exogenous serine treatment increased the mitochondrial membrane potential of degenerated strains T12 and T19 (Figure 8). At the same time, mitochondria convert the energy stored in organic matter into ATP through oxidative phosphorylation, thus providing energy for the organism [43]. In this study, it was observed that the addition of serine to the culture medium significantly increased the ATP content of degenerated strains T12 and T19 of *V. volvacea*. This increase in ATP content was likely due to serine’s contribution of one-carbon units for folate recycling during its participation in one-carbon metabolism, thereby generating more ATP for energy supply.

ROS lead to increased oxidation of unsaturated fatty acids, resulting in the production of various products such as MDA. MDA is an indicator of oxidative stress-induced membrane damage [39]. MDA, the end product of membrane lipid peroxidation, affects the respiratory chain complex of mitochondria as well as the activity of major mitochondrial enzymes. Excessive accumulation of ROS content in mitochondria leads to damage to the cellular membrane system, causing an increase in the amount of MDA [44]. Yale et al. found that the ROS content of hepatocytes under high-fat stress conditions increased, mitochondrial membrane potential decreased, ATP content decreased, and MDA content increased [45]. Mostofa et al. found that MDA content increased ROS production under high salt stress, causing oxidative stress and DNA damage [46]. This study showed that the exogenous addition of serine significantly reduced the MDA content of *V. volvacea* degenerated strains and decreased the degree of membrane damage. This change could protect the mitochondrial membrane from injury and help the recovery of *V. volvacea* degenerated strains.

*V. volvacea* is a common grass-rotting fungus, and its cultivation substrates typically comprise waste cotton, rice straw, cottonseed husk, etc., primarily composed of cellulose, hemicellulose, and lignin. *V. volvacea* secretes relevant matrix-degrading enzymes that break down substrates into smaller molecules for growth, development, and reproduction [47]. The breakdown of cellulose begins with the cleavage of long-chain cellulose into shorter oligosaccharides by EG. Subsequently, it degrades into cellobiose through CBH and ultimately converts into D-glucose with the assistance of BGL [48]. Xylan is the main hemicellulose, and Xyl can hydrolyze long-chain xylan into short-chain oligosaccharides, which further degrade into smaller molecules like xylose [49]. The degradation of lignin relies mainly on Lac and MnP, ultimately generating small molecule compounds such as phenolic monomers [50]. This study found that the exogenous addition of serine significantly increased the activities of FPA, EG, and Lac in T6, T12, and T19. Furthermore, the activities of BGL and MnP in T12 and T19 were significantly higher (Figure 11). This may be attributed to the antioxidant properties of serine, which effectively scavenge the accumulated ROS in the degenerated strains of *V. volvacea*, thereby enhancing their ability to secrete substrate-degrading enzymes [51]. This, in turn, provides nutrients such as glucose, xylose, and phenolic monomers for the *V. volvacea* mycelial, promoting mycelial growth, increasing mycelial biomass, and enhancing mycelial branching (Figure 1 and Figure 2).

In summary, this study suggests that serine can restore the production characteristics of *V. volvacea* degenerated strains through the following mechanisms (depicted in Figure 13). By providing one-carbon units to the folic acid cycle and methionine cycle in one-carbon metabolism, serine can up-regulate the gene expression of antioxidant enzyme genes related to ROS scavenging, such as *Cu/Zn-sod*, *Mn-sod2*, and gpx. Serine also increases the vitality of antioxidant enzymes, such as SOD and GPX, reduced H_2_O_2_, O_2_^−^ and MDA content, and enhances the activity of antioxidant enzymes. All of these help repair oxidative damage to cells, improving matrix-degrading enzyme activity and boosting mitochondrial function. Consequently, more ATPs promote mycelial growth, enhancing fruiting capacity and the fruiting body yield.

## 5. Conclusions

This study explored the rejuvenation of *V. volvacea* degenerated strains by the exogenous addition of serine. The results revealed that serine treatment had no significant effect on T0 and T6 but could effectively restore the production characteristics of T12 and T19 strains. Serine shortened the production cycle of T12 and restored its biological efficiency to the level of T0. Moreover, it enabled T19 to regrow its fruiting bodies. Meanwhile, exogenous serine up-regulated the expression of certain antioxidant enzyme genes, increased the activity of SOD, CAT, and GPX in T12 and T19, reduced the accumulation of ROS, lowered MDA content, enhanced MMP, and restored mitochondrial morphology. This study provides a theoretical foundation for rejuvenating and preventing degeneration in *V. volvacea* and other edible mushrooms.

## Figures and Tables

**Figure 1 jof-10-00540-f001:**
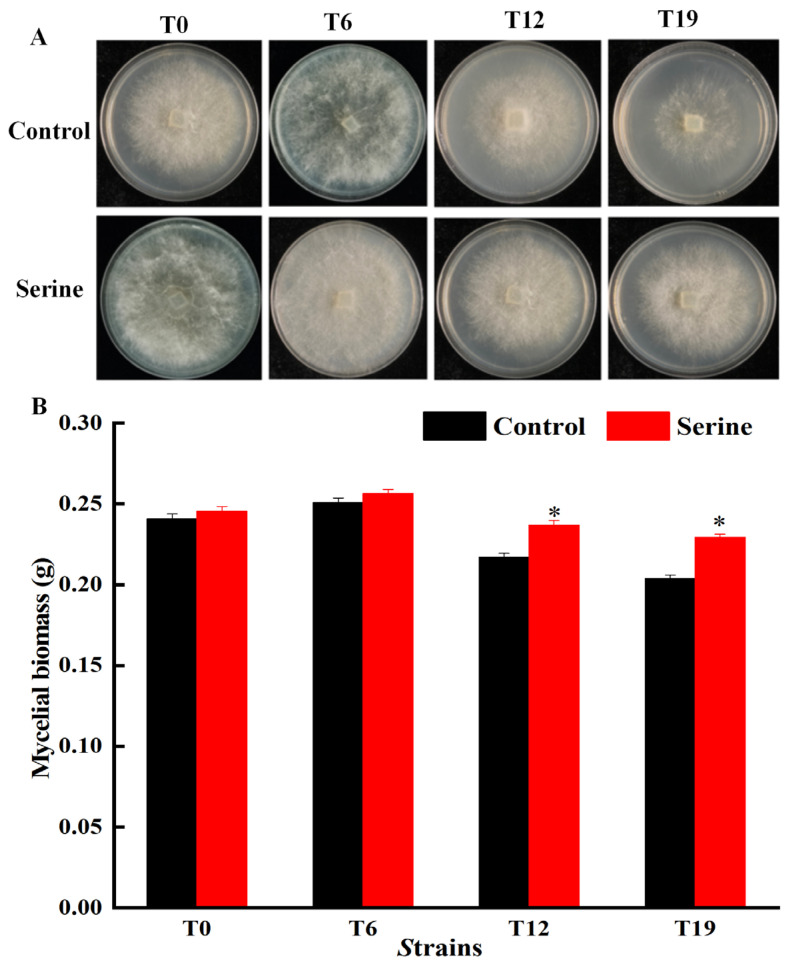
Changes of colony diameter (**A**) and mycelial biomass (**B**) of *V. volvacea*. T0 was the original strain; T6, T12, and T19 were obtained after 6, 12, and 19 successive subculture. * represents a significant difference within the same group (*p* < 0.05), the same below.

**Figure 2 jof-10-00540-f002:**
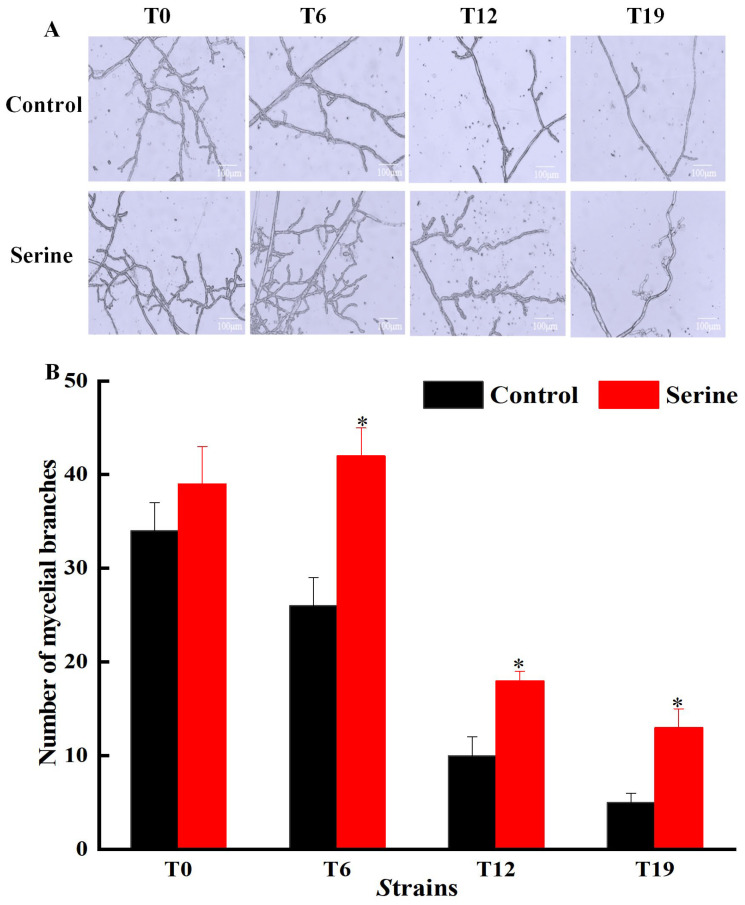
Changes of mycelial branching (**A**) and number of mycelial branches (**B**) of *V. volvacea*. * represents a significant difference within the same group (*p* < 0.05).

**Figure 3 jof-10-00540-f003:**
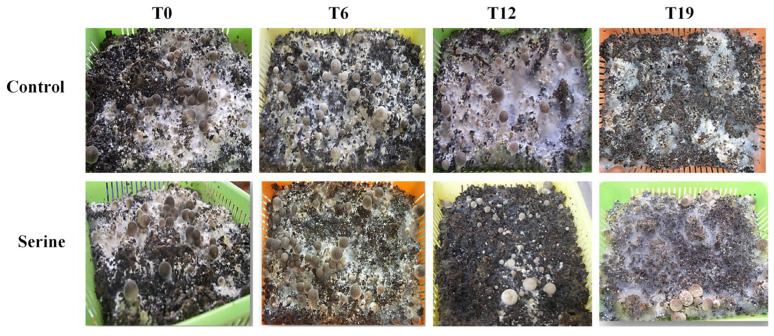
Changes in production trait of *V. volvacea*.

**Figure 4 jof-10-00540-f004:**
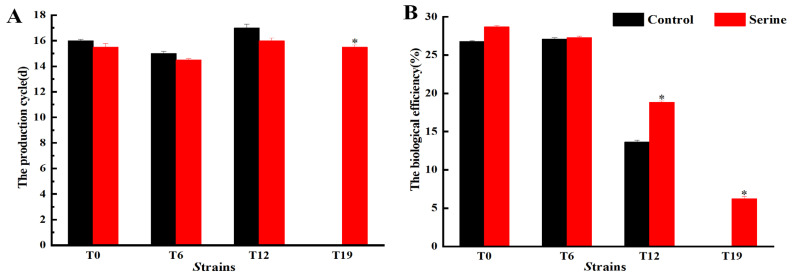
Changes in the production cycle (**A**) and biological efficiency (**B**) of *V. volvacea*. * represents a significant difference within the same group (*p* < 0.05).

**Figure 5 jof-10-00540-f005:**
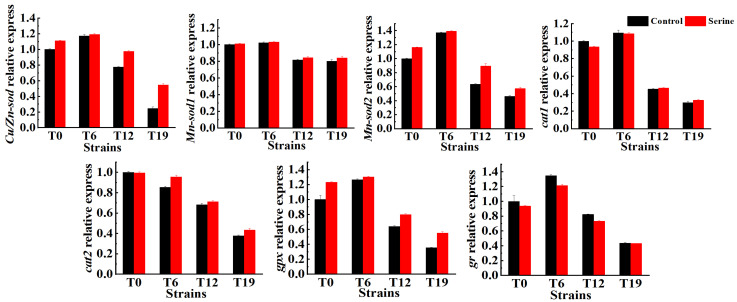
Relative expression changes of antioxidase genes in *V. volvacea*.

**Figure 6 jof-10-00540-f006:**
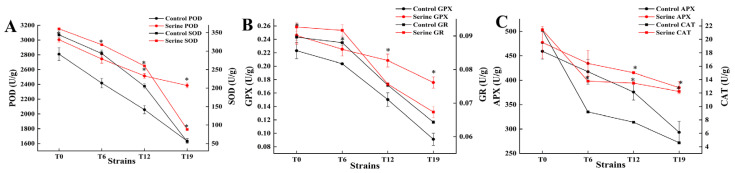
Changes of POD and SOD (**A**), GPX and GR (**B**), and APX and CAT (**C**) in *V. volvacea.* * represents a significant difference within the same group (*p* < 0.05).

**Figure 7 jof-10-00540-f007:**
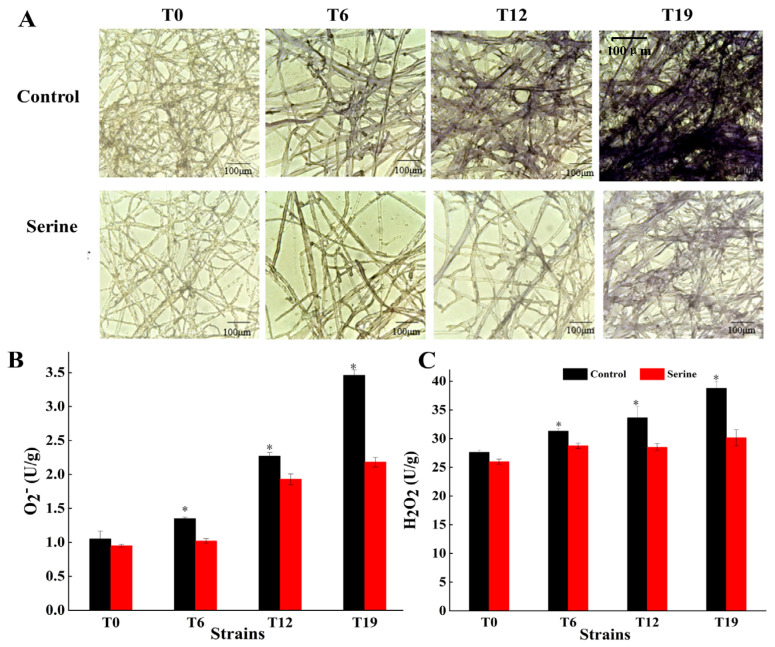
Changes of NBT staining (**A**), O_2_^−^ content (**B**), and H_2_O_2_ content (**C**) of *V. volvacea.* * represents a significant difference within the same group (*p* < 0.05).

**Figure 8 jof-10-00540-f008:**
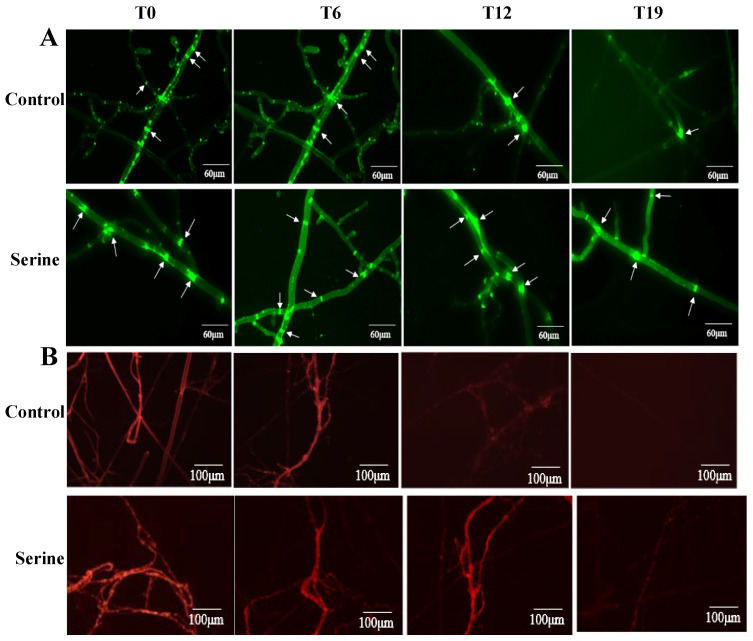
Changes of mitochondrial membrane potential (**A**) and mitochondrial morphology (**B**). Arrows indicate the appearance of bright spots in response to changes in mitochondrial morphology after specific staining of the strain.

**Figure 9 jof-10-00540-f009:**
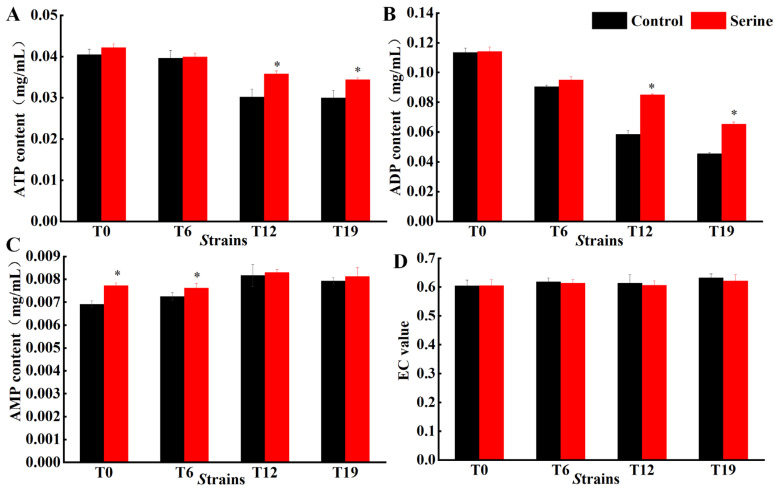
Changes of ATP (**A**), ADP (**B**), AMP contents (**C**), and EC value (**D**) in *V. volvacea.* * represents a significant difference within the same group (*p* < 0.05).

**Figure 10 jof-10-00540-f010:**
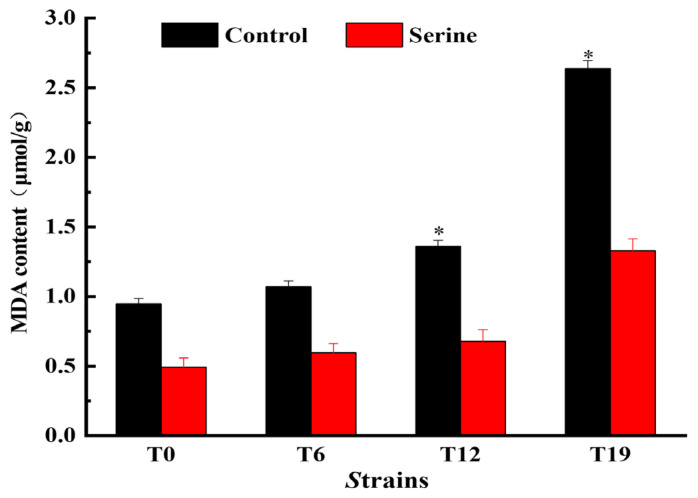
Changes in MDA content of *V. volvacea.* * represents a significant difference within the same group (*p* < 0.05).

**Figure 11 jof-10-00540-f011:**
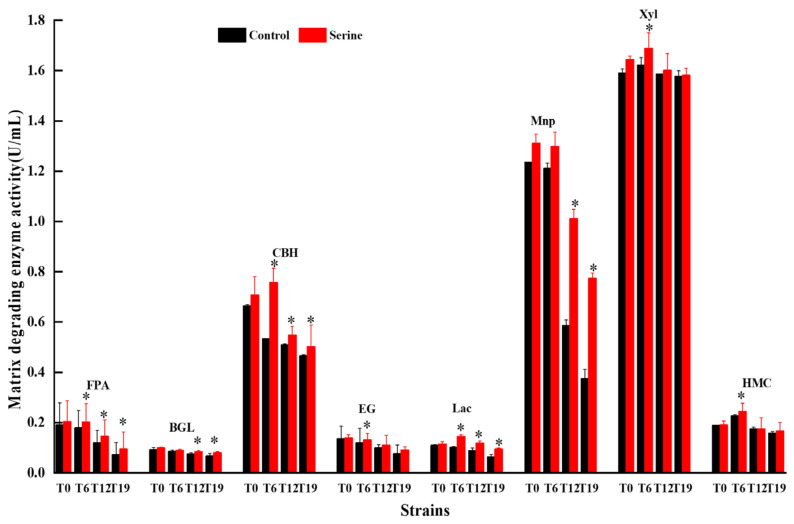
Changes in matrix-degrading enzymes activities of *V. volvacea.* * represents a significant difference within the same group (*p* < 0.05).

**Figure 12 jof-10-00540-f012:**
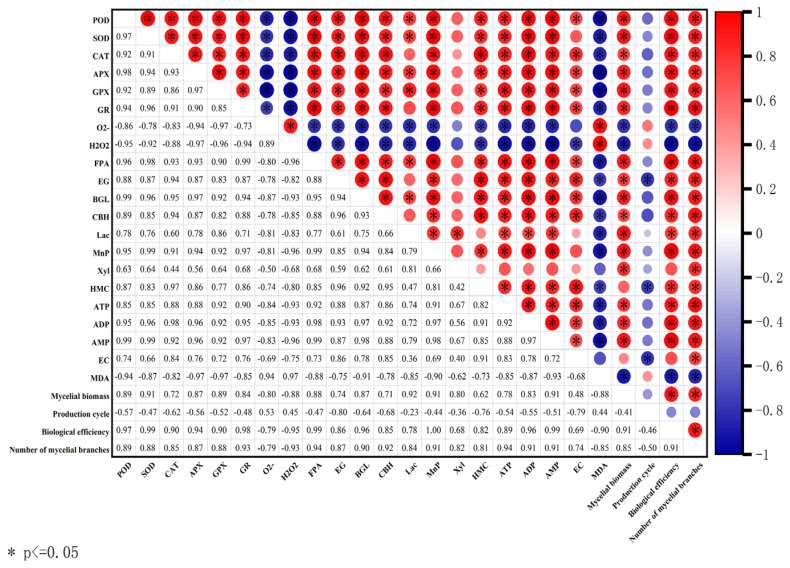
Pearson correlation coefficient matrix between each index of *V. volvacea* subcultured strains. Red and blue colors represent positive and negative correlations between factors.

**Figure 13 jof-10-00540-f013:**
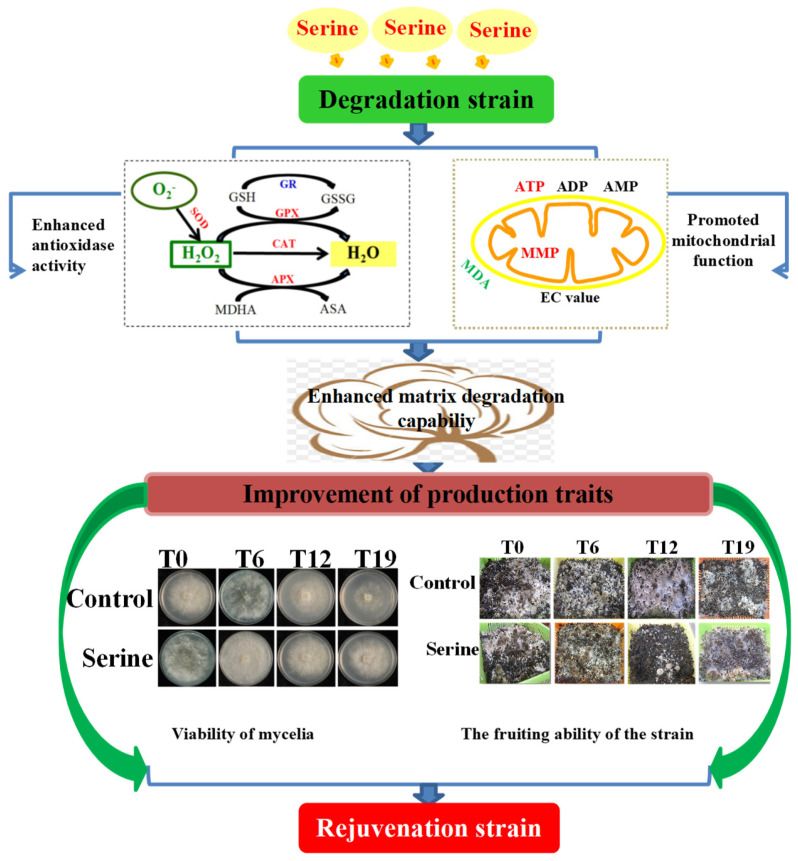
Schematic representation of degenerating strain rejuvenated by serine.

## Data Availability

The original contributions presented in the study are included in the article/Appendix A, further inquiries can be directed to the corresponding author.

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
