# Peer review of "Serine Rejuvenated Degenerated Volvariella volvacea by Enhancing ROS Scavenging Ability and Mitochondrial Function"

_jof, 2024, doi:10.3390/jof10080540_

Round 1

Reviewer 1 Report

Take into account the comments and correct the document

Figure 1 needs to describe what is meant by T0, T6, T12 , T19, and the figure has too much information.

It is recommended to separate it, as it consists of 2 figures and 2 tables in the same figure (Figure 1).

The same is true for Figure 2

Figure 3 has information that is not distinguishable due to the size, it is recommended to improve the image and increase its size to visualise it better.

In general, the results need to improve their figures.

Their discussion and conclusions are good and clear with respect to the results obtained in this work.

Author Response

We are grateful to you for comments and suggestions. Below we detail our responses to your comments and list the alterations we have made to the manuscript. Thank you very much for your valuable comments and constructive suggestions, which are vital to improve the quality of our manuscript. The revised parts are shown in red in our revised manuscript.

Comment 1: Figure 1 needs to describe what is meant by T0, T6, T12 , T19, and the figure has too much information.

It is recommended to separate it, as it consists of 2 figures and 2 tables in the same figure (Figure 1).

Response 1: Thank you for your valuable advice, the expert opinion has been adopted.

We split Figure 1 into Figure 1 and Figure 2, and enlarged the font of the axis scales and labels of the bar graphs to make them display more clearly. We labelled the meanings of T0, T6, T12, and T19 in the figure notes; see the revised manuscript.(lines215-217)

Comment 2:The same is true for Figure 2

Response 2: Thank you for your valuable advice, the expert opinion has been adopted.

We split Figure 2 into Figure 3 and Figure 4, and enlarged the font of the axes scales and labels in Figure 4 to show them more clearly; see the revised manuscript.

Comment 3:Figure 3 has information that is not distinguishable due to the size, it is recommended to improve the image and increase its size to visualise it better.

In general, the results need to improve their figures.

Their discussion and conclusions are good and clear with respect to the results obtained in this work.

Response 3: Thank you for your valuable advice, the expert opinion has been adopted.

We have enlarged the font size of the figure axis scales and labels to make the information in the images clearer; see Figures 5 and 9 in the revised version for details.

Because some pictures in the revised manuscript have been adjusted according to expert opinions, the serial number of the pictures has changed, and we have marked them in red in the revised manuscript.

Thank you so much!

Reviewer 2 Report

This paper provide in depth work regarding the use of Serine for rejuvenating the Volvariella volvacea species. The finding is interesting suggesting the potential of the Serine as rejuvenating agent in which improve the physiological characteristic of the species. It is interesting to have a further analysis whether Serine concentration might affecting the physiological characteristic of the species.

Referring line 347, please add discussion regarding the mycelial characteristic of the study in stead of starting with the biological efficiency result.

Author Response

We are grateful to you for comments and suggestions. Below we detail our responses to your comments and list the alterations we have made to the manuscript. Thank you very much for your valuable comments and constructive suggestions, which are vital to improve the quality of our manuscript. The revised parts are shown in red in our revised manuscript.

Comment 1: Referring line 347, please add discussion regarding the mycelial characteristic of the study in stead of starting with the biological efficiency result.

Response 1: Thank you for your valuable advice, the expert opinion has been adopted.

We added “Cao et al. found that under salt, dehydration, or cold stress, treatment with 5 mM GABA significantly enhanced the mycelial growth rate and density of both H. marmoreus strains by promoting front hyphae branching[28]. Our study found that the addition of serine to the culture medium had no significant effect on the mycelial and fruiting body indicators of both T0 and T6 strains. Exogenous serine significantly increased the colony diameter, mycelial biomass and mycelial branching of T12 and T19 (Figures 1,2), shortened the production cycle and significantly improved the biological efficiency of T12, restoring it to the level of T0; Even the T19 strain, which had lost fruiting ability, was capable of regrowing fruiting bodies (Figure 3)”.in “4 Discussion” of the revised manuscript.(lines367-376)

Thank you so much!
